

# Patterns and controlling factors of soil carbon sequestration in nitrogen-limited and -rich forests in China—a meta-analysis

Mbezele Junior Yannick Ngaba[1,2,3], Yves Uwiragiye[1,2,4] and Jianbin Zhou[1,2]

[1] College of Natural Resources and Environment, Northwest A&F University, Yangling, Shaanxi, China
[2] Key Laboratory of Plant Nutrition and the Agri-Environment in Northwest China, Ministry of Agriculture, Yangling, Shaanxi, China
[3] College of Resources and Environment, Southwest University, Chongqing, China
[4] University of Technology and Arts of Byumba, Byumba, Rwanda

## ABSTRACT

Soil organic carbon (SOC) management has the potential to contribute to climate change mitigation by reducing atmospheric carbon dioxide ($CO_2$). Understanding the changes in forest nitrogen (N) deposition rates has important implications for C sequestration. We explored the effects of N enrichment on soil carbon sequestration in nitrogen-limited and nitrogen-rich Chinese forests and their controlling factors. Our findings reveal that N inputs enhanced net soil C sequestration by 5.52–18.46 kg C $kg^{-1}$ N, with greater impacts in temperate forests (8.37–13.68 kg C $kg^{-1}$ N), the use of $NH_4NO_3$ fertilizer (7.78 kg $Ckg^{-1}$ N) at low N levels (<30 kg $Ckg^{-1}$ N; 9.14 kg $Ckg^{-1}$ N), and in a short period (<3 years; 12.95 kg C $kg^{-1}$ N). The nitrogen use efficiency (NUE) varied between 0.24 and 13.3 (kg C $kg^{-1}$ N) depending on the forest type and was significantly controlled by rainfall, fertilizer, and carbon-nitrogen ratio rates. Besides, N enrichment increased SOC concentration by an average of 7% and 2% for tropical and subtropical forests, respectively. Although soil carbon sequestration was higher in the topsoil compared to the subsoil, the relative influence indicated that nitrogen availability strongly impacts the SOC, followed by dissolved organic carbon concentration and mean annual precipitation. This study highlights the critical role of soil NUE processes in promoting soil C accumulation in a forest ecosystem.

## INTRODUCTION

The soil carbon pool provides for a significant portion of the total global carbon pool, accounting for more than twice the size of the terrestrial vegetation carbon pool and three times the size of the atmospheric carbon pool (*Batjes, 2014*; *Díaz-Hernández, 2010*). It contributes the most to carbon cycles and processes of any terrestrial biome, such as carbon (C) sequestration. Soil carbon sequestration is the removal of carbon dioxide gas ($CO_2$) from the atmosphere (*i.e.,* photosynthesis) by natural or artificial means

Corresponding authors
Mbezele Junior Yannick Ngaba,
ngabajunior@yahoo.fr
Jianbin Zhou, jbzhou@nwsuaf.edu.cn

(*Zhang et al., 2021*). The photosynthetic process of trees and plants, which stores carbon as they extract $CO_2$ from the atmosphere, is the most widespread example in nature (*Law, 2013*). Subsequently, forest soil C sequestration mitigates global climate change by reducing greenhouse gas emissions and protecting C sequestration (*Khorchani et al., 2022*; *Li et al., 2022a*). Several biological factors influence the processes of soil organic carbon sequestration and decomposition, such as soil depth, temperature sensitivity (*Hyvönen et al., 2007*; *Xu, Luo & Zhou, 2012*), N deposition (*Law, 2013*; *Schulte-Uebbing, Ros & De Vries, 2022*), climatic conditions (*Liu, Shao & Wang, 2011*; *Ngaba et al., 2019*), soil aggregate distribution (*Liu et al., 2018*; *Ngaba, Bol & Hu, 2021*). Further, soil organic carbon (SOC) sequestration significantly decreases with land use change (*Arunrat et al., 2022*; *Ngaba et al., 2019*), soil layers, and land use type (*Jobbágy & Jackson, 2000*; *Zhao et al., 2017*).

However, forest growth changes are mainly induced by N input, which in turn will affect soil C accumulation. For example, according to *Fleischer et al. (2013)*, increasing atmospheric nitrogen deposition in forests increases photosynthesis in boreal and temperate evergreen forests but levels off when a threshold value of 8 kilos of nitrogen per hectare per year is achieved. This long-term atmospheric N deposition will affect ecosystem productivity by increasing bioavailable N (*Zhu et al., 2021*). The main mechanisms driving soil C gains following N input in N-limited forests are plant growth stimulation and changes in fungal community structure and functioning (*Tian et al., 2019*). Recent studies have well established that soil carbon sequestration in Chinese forest ecosystems is enhanced by low levels of nitrogen and additions for a short time (*Ngaba et al., 2022*). These findings underscore the need to understand the links between the C and N cycles in the ecosystem.

In fact, nitrogen (N) deposition is a component of global change that considerably affects plant productivity and carbon sequestration in forest ecosystems, including those in China. It can enhance C sequestration by increasing net primary production (NPP) and decreasing soil organic carbon decomposition (*Schulte-Uebbing, Ros & De Vries, 2022*). As long as N deposition occurs, NPP is stimulated until N saturation is reached, increasing litter inputs both above and belowground (*Schulte-Uebbing & De Vries, 2018*). The breaking down of dead plants and organic matter (OM) in the soil (*Frey et al., 2014*; *Zak et al., 2017*), the inhibition of soil respiration (*Liu & Greaver, 2010*), or the alteration of microbial enzymatic activity (*DeForest et al., 2004*) have all been shown to promote soil C sequestration (*DeForest et al., 2004*; *Frey et al., 2014*). Consequently, assessing nitrogen use efficiency (NUE) is crucial because it could acidify the soil and accelerate soil nutrient loss (*Tang et al., 2022*). Besides, N availability reduced the fungal/bacterial ratios in the forest soils and increased the biomass of bacteria (*Cusack et al., 2011*; *Zhou et al., 2017*). Hence, N enrichment may directly affect C sequestration by gaining or releasing carbon from the soil. Therefore, there are many gaps in our knowledge of how N deposition in forests affects soil C sequestration in China.

The first limitation, they focus more on the region where it has been well reported that there has been a decrease in N deposition (*Hole & Engardt, 2008*) rather than focusing on the hot spot of N sequestration like China, which is the world's leading N producer and emitter (*Liu et al., 2013*). Human activities in Asia have heavily influenced the natural

nitrogen cycle, particularly in China. In the last few decades, China's rapid economic expansion has led to significant levels of nitrogen emissions (*Gao et al., 2020*). Human activities aggravate nitrogen-deposition pollution in China by increasing atmospheric N deposition from fossil fuel combustion and fertilizing croplands with nitrogen (*Galloway et al., 2004*; *Gao et al., 2020*). Nitrogen deposition in China increased from 13.2 kg ha$^{-1}$ yr$^{-1}$ in the 1980s to 21.1 kg ha$^{-1}$ yr$^{-1}$ in the 2000s because human-caused additions of nitrogen to the atmosphere over time and space as a result of agricultural fertilizer use and fossil-fuel combustion (*Law, 2013*). These activities have also changed the environment, such as the climate, which can affect the metabolic rates of biological processes, such as microbial-mediated transformations of organic N compounds to inorganic N (*i.e.,* N mineralization) associated with the decomposition of soil organic matter (SOM) and biological N fixation, which could make more N available (*An et al., 2022*; *Poeplau et al., 2018*; *Yu & Zhuang, 2020*). Environmental and human health could benefit or be damaged as N deposition increases on the Chinese surface. Indeed, once emitted into the air, ammonia and nitrogen oxides, two important nitrogen pollutants, can be converted to secondary pollutants like ammonium and nitrates in the atmosphere and then carried back to earth by rain and snow, a process known as N deposition (*Liu et al., 2022*; *Yu et al., 2019*). The atmospheric deposition of N is a significant effect of pollution. It may have a beneficial or detrimental impact on forest ecosystems, for example, by increasing tree growth or causing acidification, eutrophication, or soil cation degradation (*Marchetto et al., 2021*). Thus, how N deposition will impact C sequestration in Chinese forests remains uncertain due to the spatial and temporal patterns of changes.

Secondly, recent studies on forest soil C sequestration have overwhelmingly emphasized the impact of chronic N addition on C gain in boreal forests (*Fleischer et al., 2015*; *Hoegberg et al., 2006*; *Hyvönen et al., 2007*; *Magnani et al., 2007*) and tropical forests (*Baccini et al., 2017*; *Cusack et al., 2011*; *Lu et al., 2021*; *Tian et al., 2019*). Meanwhile, N deposition also significantly impacts several mechanisms in temperate and tropical forests, which may have a critical role in the changing atmospheric carbon concentrations. Changes in fungal community composition and function in temperate forest ecosystems, which are frequently N-limited in nature, were the primary mechanisms for increasing soil C following the enrichment of N (*Hassett et al., 2009*; *Hesse et al., 2015*). Nitrogen deficiency is widespread and common in temperate forests, and it decreases stand productivity by reducing leaf area, light interception, and photosynthesis to a lesser extent (*Reich et al., 2008*).

Yet, N deposition generally has a minor effect on plant growth in N-rich subtropical forests; however, plant growth stimulation and changes in the composition and functions of the fungal community are the main factors that lead to soil C gains after N deposition in N-limited forests (*Tian et al., 2019*). Also, subtropical forests differ from temperate forests not only in climate, for example, subtropical forest soils are warmer than temperate, but also in that they are more phosphorus (P) limited than N restricted, and the soils are frequently extremely acidic with low base cation concentrations (*Hall & Matson, 2003*). So, the relative soil C sequestration in temperate and subtropical forest ecosystems may respond differently to N status (N-rich or N-limited) through atmospheric deposition. It

is therefore critical to improve our understanding and accurately quantify the net effect of N enrichment on soil C sequestration in China to prevent climate change. The specific objectives of this study are to (1) figure out how C sequestration responds to N addition with forest type, N fertilization rate, and duration; (2) to investigate the relationship between soil C sequestration and controlling factors, as well as their relative importance under N enrichment; and (3) determine how climate factors influence soil C sequestration patterns.

## MATERIALS AND METHODS

### Study sites description

China has a wide variety of climates due to its vast size. Summers are hot everywhere except the highlands and high mountains, and winters are severely cold in the north, mountains, and plateaus. Summer is the wettest season, with the exception of the vast desert areas in the West, where precipitation is limited and erratic (*Tian et al., 2019*). The southeast receives the greatest precipitation, while the northwest receives the least. The average annual temperature is 21 °C (range from 13 °C in January to 28 °C in July), and the climate is monsoonal humid tropical, with 1.930 mm of precipitation falling in a clear seasonal pattern (75% from March to August, and 6% from December to February). The soil type is lateritic red earth produced from sandstone (Oxisol) (*Tian et al., 2019*). Forests cover around 22% of China's total land area. China has a total forest area of around 206.861.000 hectares, with approximately 5.6% (or 11.632.000 hectares) designated as primary forest. Between 1990 and 2010, China increased around 49.720.000 hectares (31.6%) of forest cover (*Qin, Huang & Luo, 2010*). Huge swathes of southern China are covered with forest, stretching from Fujian Province in the east to Sichuan and Yunnan Provinces in the west. Much of China's far northeast is likewise forested. Forest cover declines considerably between the densely populated cities of Shanghai and Beijing and the distant western provinces of Xinjiang and Tibet (*Zhu, 2017*).

### Data collection

We searched the Web of Science and Google Scholar for peer-reviewed journal publications published between 2000 and 2022 that investigated the patterns and controlling factors of soil carbon sequestration in Chinese forest (Table S1, Fig. S1). This period was chosen because it corresponds to the start date of the Gain for Green project (GFGP) or the Natural Forest Protection Program. We utilized terms like "patterns and controlling factors", "soil carbon sequestration", "nitrogen-limited and nitrogen-rich forests", "nitrogen enrichment", "nitrogen deposition", "nitrogen addition", and "nitrogen fertilization" as well as the keywords "forest" and "carbon". Overall, 1,203 papers were obtained for this study, but only 61 were retained following the screening stage to exclude irrelevant publications (Fig. 1). There were eight variables in the database notably forest types, N addition forms ($NH_4NO_3$, Urea), N addition rates (30, 30–70, and >70 kg N ha$^{-1}$ yr$^{-1}$), N deposition rates (*i.e.,* 10, 10–20, and >20 kg N ha$^{-1}$ yr$^{-1}$), N fertilization durations (1, 1–3, and >3 years), and N deposition rates were all compared. Soil depths (cm) ranged from 0 to 10, 10 to 20, and 20 to 40 cm, with experimental durations ranging from 6 to
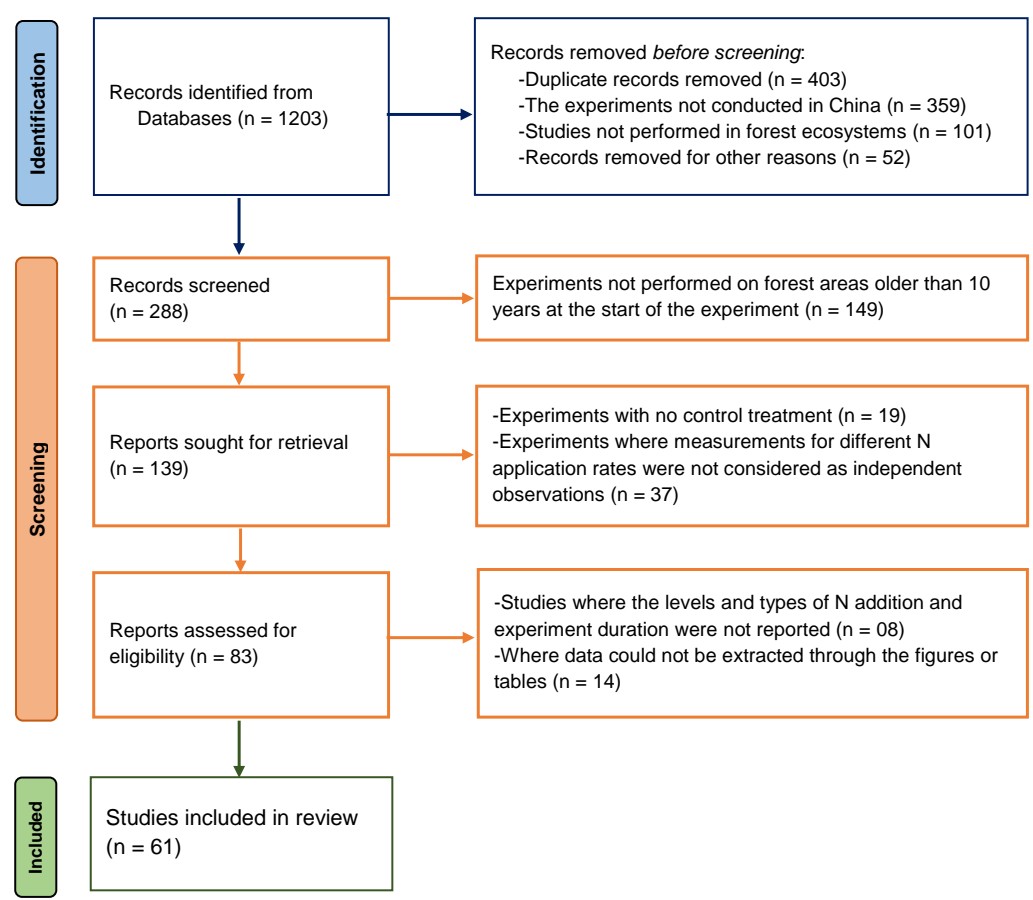

**Figure 1 Flow diagram for inclusion and exclusion of Studies.**

15, 15 to 35, and >35 years. Data was collected as previously described in Supplement File and *Ngaba et al. (2022)*.

## Analysis and statistics

The meta-analysis approach outlined by *Hedges, Gurevitch & Curtis (1999)* was used to evaluate the patterns and controlling variables of soil carbon sequestration based on the natural log-transformed response ratio (RR) using the METAWIN version 2.1 (Sinauer Associates, Inc., Sunderland, MA, USA). In addition, we calculated the relative influence technique to calculate the relative importance of predictor factors of soil C sequestration (independent variables). The formulas applied has been used in previous works (*Ngaba et al., 2022*). Statistical analyses and graphs were determined with OriginPro 2021 and SPSS 26.0 software (SPSS, Inc., Chicago, IL, USA). Correlations were considered significant at $p < 0.01$ and $p < 0.05$.
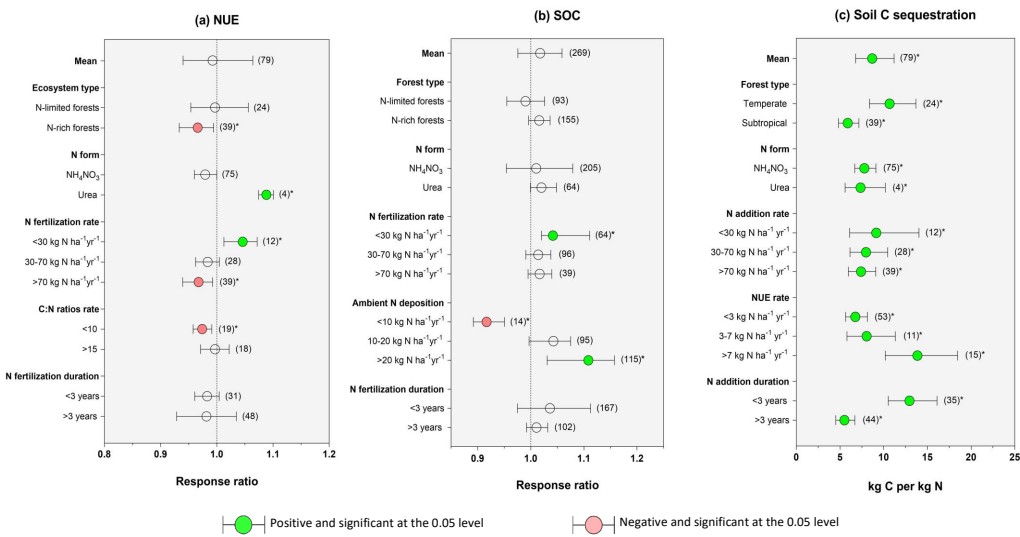

**Figure 2** Effects of N addition on (A) nitrogen use efficiency, (B) soil carbon organic carbon concentration, and (C) soil C sequestration in temperate and subtropical forests of China. Black bars represent 95% confidence intervals. The different letters in parentheses represent the number of observations. The dashed vertical line was drawn at a mean effect size of 1. Green color indicates correlation is positive and significant at the 0.05 level; Red color indicates correlation is negative and significant at the 0.05 level, and an asterisk (*) indicates that the correlation is significant at the 0.05 level. NUE, nitrogen use efficiency; SOC, soil carbon organic carbon concentration.

# RESULTS

## Effect of N enrichment soil N use efficiency (NUE) and soil organic carbon (SOC) sequestration

In our meta-analysis, N addition significantly affected NUE ($P < 0.05$; Fig. 2A). The estimates of NUE varied between 0.24 and 13.3 kg C kg$^{-1}$ N depending on the forest type. Furthermore, NUE response to N addition decreased significantly in subtropical forests ($P < 0.05$), N form (NH$_4$NO$_3$), high C:N ratios ($>15$), and long-term application ($>3$ years) (all, $P < 0.01$, Fig. 2A). At the same time, it significantly increased in low application rate ($P < 0.05$). On the other hand, there was a positive relation between NUE and C/N ratio response after adding N fertilizer but had no significant effect (Fig. 2A). On the other hand, N addition increased soil organic carbon sequestration by an average of 6%, but the effect was insignificant (Fig. 2B). The enrichment was positive for tropical and subtropical forests (+7% and +2%, respectively). However, when the data were separated into low N fertilization rate (30 kg N ha$^{-1}$yr$^{-1}$; +2%) and high ambient N deposition rate ($>20$ kg N ha$^{-1}$yr$^{-1}$; +12%), N addition had a positive and significant effect (Fig. 2B; $P < 0.05$). The nitrogen use efficiency and mean annual precipitation (MAP) have significant linear relationships ($R^2 = 0.34$, $P < 0.01$) (Fig. S2).

## Patterns of soil C sequestration in response to N addition

The results showed that N addition significantly increased C sequestration across Chinese forest ecosystems ($P < 0.05$; Fig. 2C), with values ranging between 6.8 and 11.2 kg C

kg$^{-1}$ N (Fig. 2C). Also, C sequestration response was significantly higher in temperate forests (10.6:8.37–13.68 kg C kg$^{-1}$ N) compared to subtropical forests (5.87:4.83–7.16 kg C kg$^{-1}$ N). Figure 2C also highlights that C sequestration was dependent on N form, N addition levels, ($R^2 = 0.34$, $P < 0.01$) rate, and N application duration ($P < 0.05$; Fig. 2C). In particular, this study showed that NH$_4$NO$_3$ form, low N rates, high NUE levels, and short-time N application strongly influenced C storage processes (Fig. 2C).

### Soil C sequestration in response to N addition rates and duration
The rate of fertilization influenced the responses of the C dynamic to N addition. In general, a low application rate (<30 kg N ha$^{-1}$yr$^{-1}$) did not significantly increase soil C concentration in Chinese forests (+4%). While high application rates (>70 kg N ha$^{-1}$yr$^{-1}$) reduce soil C concentration by 7% ($P < 0.01$), especially in NH$_4$NO$_3$ form (−9%) and long-term application (>3 years, −2%) (Fig. 3A; $P < 0.01$). However, C accumulation was more pronounced in temperate forests (−3%) compared to subtropical (−7%) (Fig. 3A, $P < 0.01$). When the data was subdivided by N application duration, we found that short-term application increased soil C concentration in Chinese forests by +5% (Fig. 3B; $P < 0.01$), while long-term application decreased by −10% (Fig. 3B; $P < 0.01$).

### Controlling factors of soil C sequestration
Overall, we found a negative correlation between SOC concentration, bulk density, pH, and sand content ($P < 0.01$; $R^2 = -0.59$, $R^2 = -0.94$, $R^2 = -0.58$ for Figs. 4A, 4D and 4F, respectively). While it was positive with total nitrogen (TN), dissolved organic carbon concentration (DOC), clay, silt content, and soil aggregate size ($R^2 = 0.94$, $R^2 = 0.94$, $R^2 = 0.13$, $R^2 = 0.67$, $R^2 = 0.66$ for Figs. 4B, 4C, 4E, 4G and 4H, respectively). However, SOC concentration was higher in macro-aggregates than in micro-aggregates (Fig. S3). Our meta-analysis showed that although soil carbon concentration varied with season types, it increased with forest age and elevation but decreased with soil depth layers (Fig. 5). Concerning climate factors, SOC concentration had a negative and significant relationship with mean annual temperature (MAT) but a positive relationship with mean annual precipitation (MAP) depending on their magnitude ($P = 0.91$, Fig. S4). In addition, the relative influence analysis indicated that N availability (40%) is one of the major factors controlling C accumulation, followed by DOC > MAP > MAT > clay > pH > sand > silt (Fig. 6).

## DISCUSSION

### Ecosystem types, N addition levels, and duration drive C sequestration
Understanding the changes in forest nitrogen (N) deposition rates has important implications for C sequestration. Our meta-analysis showed that N addition significantly increased soil C sequestration in the temperate forest compared to the subtropical forest (Fig. 3), as *Fleischer et al. (2015)* demonstrated. This could result in how N availability relates to forest productivity because ecosystem productivity and species composition vary widely in China as a result of regional variations in temperature, precipitation, and

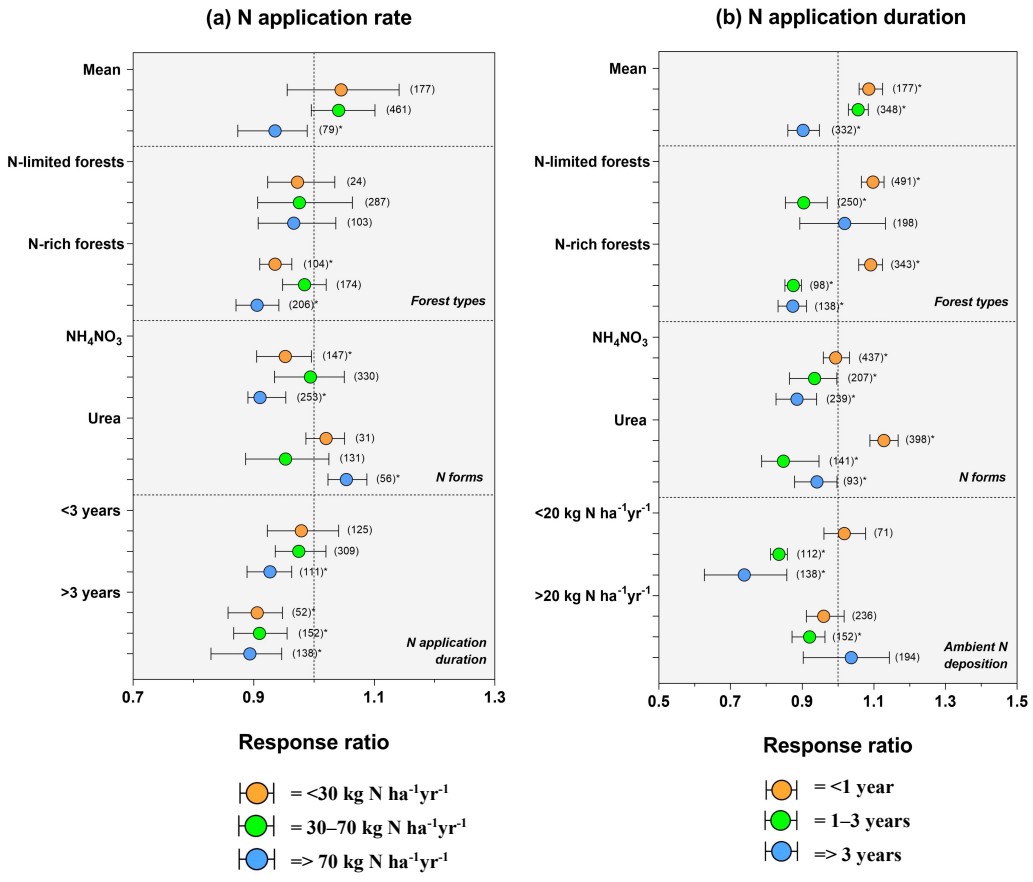

**Figure 3** Soil C concentration response to (A) N application rate and (B) N application duration in N-limited (temperate) and N-rich forests (subtropical) of China. Black bars represent 95% confidence intervals. The different letters in parentheses represent the number of observations. The dashed vertical line was drawn at a mean effect size of 1. *Correlation is significant at the 0.05 level.

other physiographic factors. For instance, forest productivity is determined by the amount of N available in the soil for plant use (*Binkley & Fisher, 2019*); thus, high rates of N availability result in rapid forest development. Evidence of this trend can be seen in Fig. 2; soil C sequestration increases with the level of NUE. According to *Zhu et al. (2021)*, NUE induced by N addition in Chinese terrestrial ecosystems is mainly influenced by the aridity index and precipitation.

In comparison, a gradual decrease in nitrogen deposition in the future will cause a drop in the net carbon sequestration. In general, the sink for soil carbon sequestration is often linked to plant productivity and aboveground litter inputs, which is the case in temperate forests (*Lu et al., 2021*). The low sequestration rate observed in subtropical forests can result from excess N accumulation and N leaching in the soil because soil microorganisms do not assimilate inorganic excess N added. According to *Matson, Lohse & Hall (2002)* and *Wu et al. (2022)*, over-enrichment with N can increase N loss *via* trace-gas emissions (*e.g.*, denitrification, volatilization) and solution leaching. As a result of the loss of N inputs,

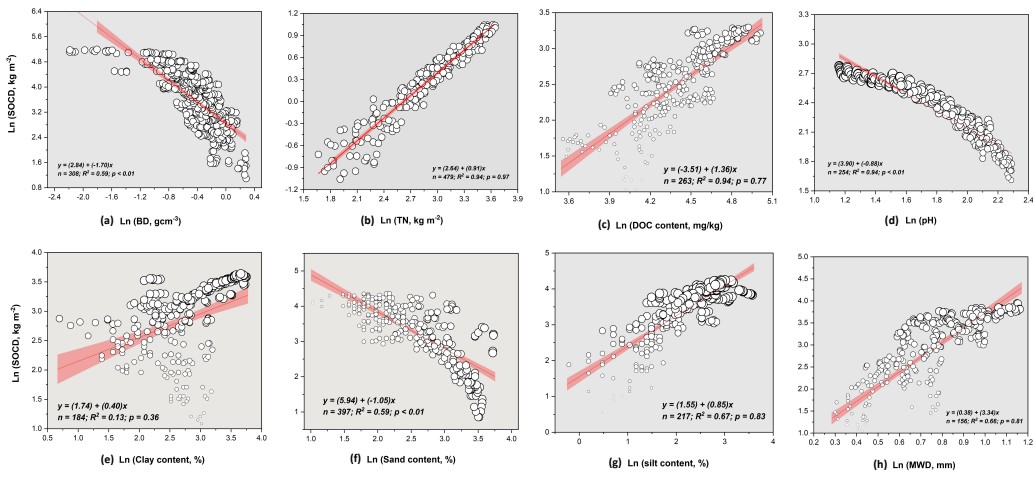

**Figure 4** **Relationship of soil carbon concentration with (A) bulk density (BD), (B) total nitrogen (TN), (C) dissolved organic carbon (DOC), (D) soil pH, (E) clay content, (F) sand content, (G) silt content, and (H) MWD under N enrichment.** The red area around the regression line represents the 95% confidence interval, whereas N is the number of observations.

NUE will decrease and lead to low N retention rate fractions in subtropical ecosystems (*Bai, Houlton & Wang, 2012*).

On the other hand, forest productivity increases when increasing water availability is involved. Temperatures and rainfall in China tend to follow the same pattern from southeast to northwest, with higher rates in subtropical forests than in temperate forests (*Li et al., 2022b*). Together, the high availability of carbon in N-rich forests and the high water supply could induce high nutrient uptake by the rhizosphere. This process is aided by nutrient recycling, which converts organic to plant-available mineral forms; nutrient storage in SOM; and physical and chemical processes that regulate nutrient sorption, availability, displacement, and eventual losses to the atmosphere and water (*e.g.*, nitrogen fixation) (*Canton, 2021*). These conditions favor faster decomposition and less SOM accumulation (*Cusack et al., 2011*) in the soil and, consequently, soil C sequestration (*Frey et al., 2014*).

Our meta-analysis reported that soil C sequestration was less pronounced in soils with high N fertilization rates ($>70$ kg N ha$^{-1}$ yr$^{-1}$) (Fig. 3), implying that soil C accumulation efficiency declines with N addition added (*Hyvönen et al., 2007*). Reduced C allocation due to N enrichment may be due to root physiology, exudation modification, and rhizosphere priming effects (*Janssens et al., 2010*; *Kuzyakov, 2002*). Considering all available evidence, it appears probable that elevated N addition decreased loss of soil C in temperate forests. Sustainable forest management practices could store the most carbon and keep the quality and productivity of the forest ecosystem (*Adam Langley, Chapman & Hungate, 2006*; *Rillig & Allen, 1999*) and result in positive or negative effects on plant growth. Several mechanisms could explain soil carbon gains following N deposition in N-limited temperate forests: the effect of N enrichment on soil acidification and mineralization of soil organic C by microorganisms by influencing nutrient demand (*Lu et al., 2021*); plant growth

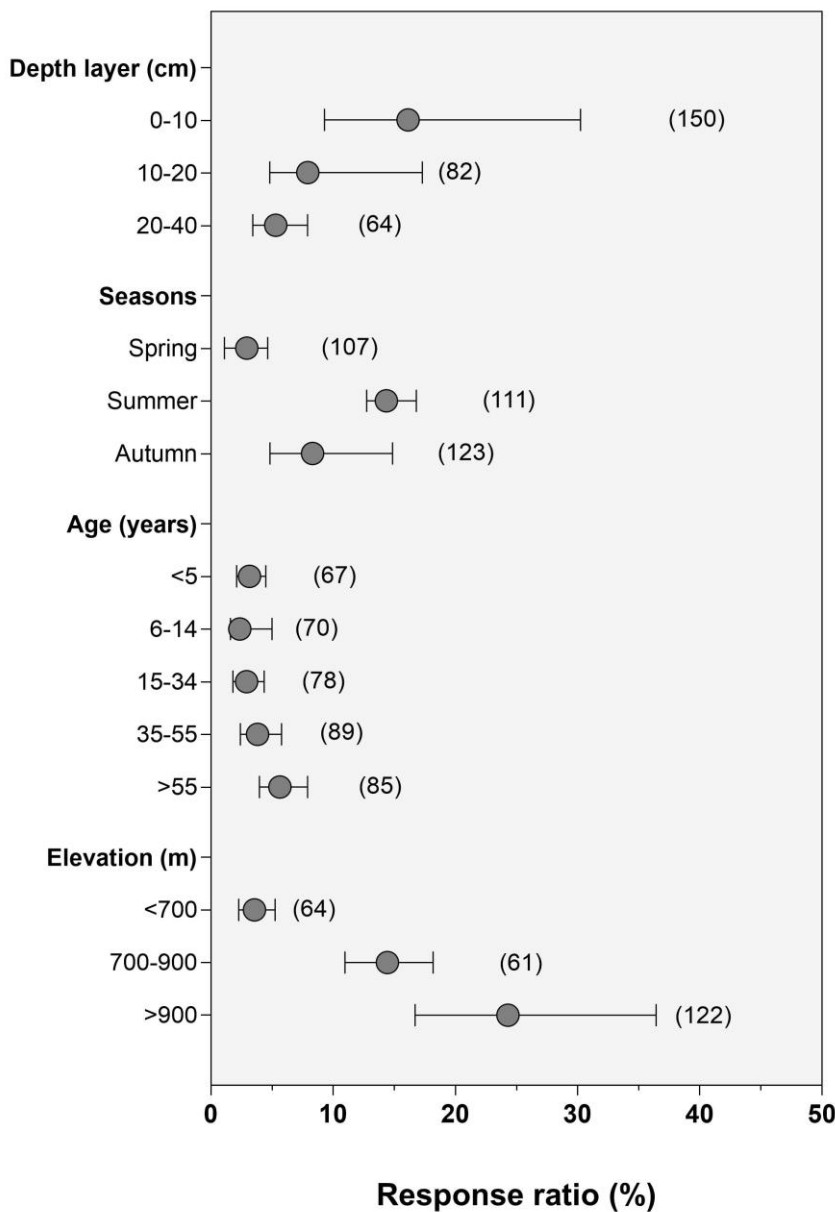

**Figure 5** Soil carbon organic carbon concentration response to N enrichment following soil depth layers, season types, age, and elevation magnitude.

stimulation, as well as shifts in the composition and functions of the fungal community (*Tian et al., 2019*). For instance, acidic deposition is known to influence the growth of plants, and the quality of soil suffers as a result (*De Vries et al., 2009*). A reduction in soil pH promotes an increase in forest floor thickness at numerous locations since roots are unable to penetrate the acidic mineral soil, and organic residues decompose more slowly on the soil surface than in the soil (*Nieder & Benbi, 2008*). Therefore, soil acidity changes will alter the equilibrium between the metabolism of the soil microorganisms and plant roots.

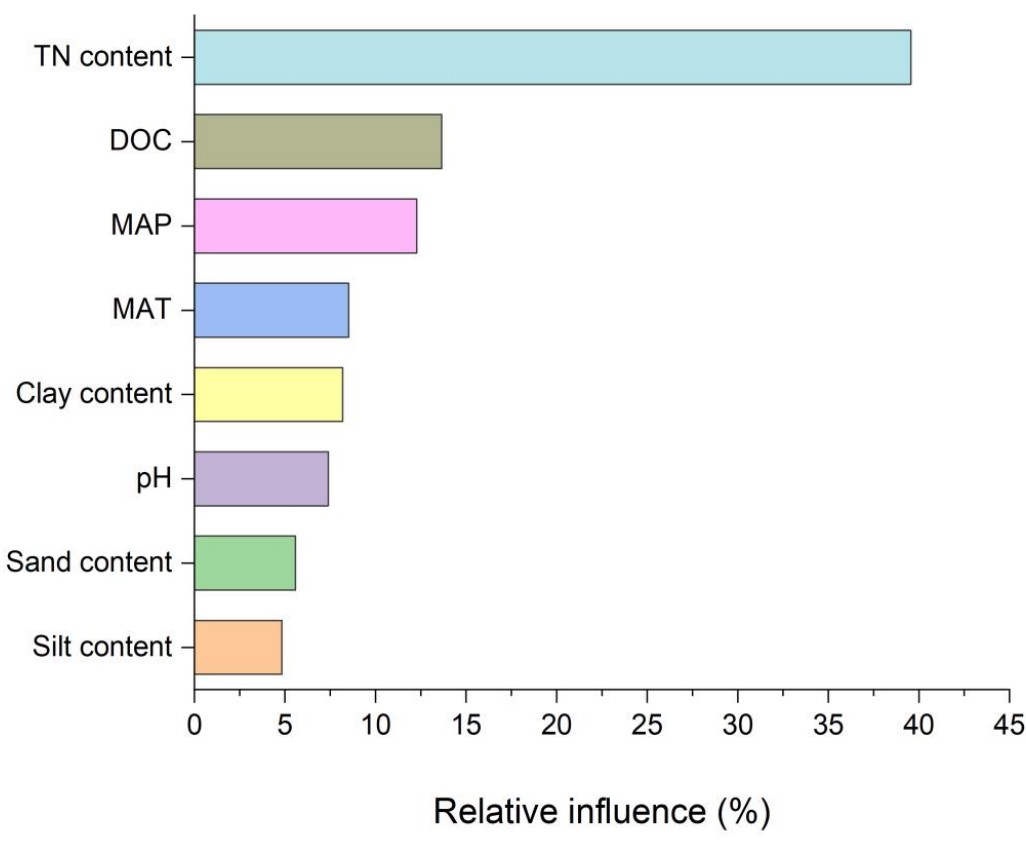

**Figure 6  Relative influence of predictors on soil organic carbon concentration.** The relative influence is proportional to the bar length. Nitrogen availability has the strongest impact on the SOC, followed by DOC, MAP, MAT, clay content, pH, sand, and silt content. The sum of relative influence is scaled to 100. TN stands for total nitrogen; DOC stands for dissolved organic carbon; MAP stands for mean annual precipitation; and MAT stands for mean annual temperature.

Considering all evidence findings, it appears probable that elevated N addition decreased loss of soil C in temperate forests. Sustainable forest management practices could store the most carbon and keep the quality and productivity of the forest ecosystem (*Diao et al., 2022*; *Xue, Kimberley & McKinley, 2022*).

Overall, our results showed that short-term N enrichment enhanced soil C sequestration, implying that microbial composition and function activities enhance fast carbon cycling. However, the evidence was only shown in the temperate sequestration rate. Our findings contradict those found by *Tian et al. (2019)*. The study revealed that high-and long-term N-enrichment resulted in an increase in soil organic carbon sequestration due to the reduction in relative abundances of bacteria and genes that degrade cellulose and chitin. The large discrepancy suggests that (i) N forest status may influence the response of C allocation to belowground under N input; (ii) fungal and bacterial responses differ following specific site conditions. Indeed, there is a long debate about whether N addition increases or decreases bacterial biomass in subtropical forest soils, fungal/bacterial ratios,

and its significant effect on complex organic SOC compounds (*Wang, Liu & Bai, 2018*; *Zhu et al., 2014*). Moreover, forest productivity is stimulated by N input until N saturation is reached. The response of C sequestration to N enrichment is also a function of the other N input sources. It has been well demonstrated that N input in soil has three major sources: atmospheric N deposition, weathering of soil minerals, and, for nitrogen, biological N fixation (*Binkley & Fisher, 2019*). The sequestration of carbon in soils depends on a number of abiotic or biotic factors.

## Abiotic and biotic controlling factors and implications for soil C sequestration

According to the hierarchy concept, soil aggregate-related OC regulates SOM dynamics, which is the primary agent for soil aggregation (*Six & Paustian, 2014*). Likewise, determining SOC sequestration in different sized aggregates helps evaluate soil quality or degradation of soils. In general, the accumulation of SOC is mainly attributed to the physical protection of OM (and thus OC and N) from environmental and microbial attack or mineralization (*Six & Paustian, 2014*). Previous studies have shown that SOC sequestration in the forest ecosystem strongly correlates with soil aggregate stability, as reflected by the MWD in this study (*He et al., 2021*). Similarly, our findings showed that SOC was significantly higher in macroaggregates than in microaggregates in Chinese forest types because of the OM binding microaggregates into macroaggregates (*Li et al., 2020*). This result suggests a slower turnover rate and lower decomposability of SOM in macroaggregates (*Ngaba, Bol & Hu, 2021*). It highlights the high rate of root decomposition within macroaggregates, increasing C sequestration. According to *Liu et al. (2018)*, macroaggregates contain more C and young labile C than microaggregates because the new input OC is first associated with microaggregates and then incorporated into macroaggregates. However, our findings contradict those found by *He et al. (2021)*. This difference could be attributed to the surface area size and the binding agents because soil microorganisms could not break down and utilize the OM associated with the micro-aggregates. For example, *Six et al. (2004)* argued that the increased specific surface area of soil micro-aggregates might improve OM adsorption from root exudates and litter residues.

Soil depth, season changes, time, and elevation were among the most important controlling factors of SOC sequestration in the Chinese forest (Figs. 5 and 6). Generally, SOC sequestration decreased with soil depth in all aggregate fractions. Our study fully agrees with this trend. We found that the pattern of SOC decreased gradually with the increase of soil depth layers, indicating that climate effects are predominantly in the topsoil (close to the surface, 0–20 cm). This decrease could be attributed to the reduction of environmental impacts in deep soil layers through increasing soil buffering ability (*Yang et al., 2010*); the variation of plant litter C input (both the surface litter and below the fine ground root) (*Tang et al., 2011*) or the slower-cycling SOC in deep soil layers (*Jobbágy & Jackson, 2000*). Climate gradients and altitude influenced SOC distribution by affecting litter generation and breakdown processes (*Ngaba et al., 2022*; *Zhao et al., 2020*). However,

tree species and forest age are two key variables affecting carbon sequestration in forest ecosystems through their impact on the balance between C input and output.

In agreement with *Yu et al. (2014)*, our study found a negative association between SOC and bulk density (BD), pH, and sand content (Fig. 4). This outcome can be explained by numerous causes, including increased soil porosity, (ii) root growth, and (iv) soil aggregate stability (*Liu, Shao & Wang, 2011*; *Ngaba, Bol & Hu, 2021*; *Yu et al., 2014*; *Liu, Shao & Wang, 2011*; *Ngaba, Bol & Hu, 2021*; *Yu et al., 2014*). On the other hand, their combined influence has a direct effect on SOC. According to *Xiao (2015)*, soil aeration pattern is determined by the interactive influence of clay content and texture, total soil porosity, or soil bulk density, which affects soil microbial respiration. Increases in soil BD, for example, have been shown to be deleterious to tree growth due to the way they alter other soil properties. As a result, soil C accumulation is often negatively related to BD, possibly due to greater rooting (root growth) in improved systems. This trend has highlighted the importance of soil BD for tree growth and, by extension, C sequestration as illustrated by *Binkley & Fisher (2019)* in a savanna region of central Brazil. Because of its effect on soil water and oxygen availability, the bulk density of the soil influences SOC decomposition *via* mineralization (*Baldock & Skjemstad, 1999*).

The processes of decomposition and nitrification are regulated by the pH of the soil (*Jackson et al., 2005*; *Zhou et al., 2019*). Soil microorganism activity is dependent on the structure and diversity of the soil microbial population. The breakdown processes of SOM were considerably influenced by soil pH, resulting in a negative correlation between SOC and pH (*Jackson et al., 2005*). Soil organic matter buildup, in particular, would accelerate the rate of soil acidification. This finding suggests that the buildup of SOC in the soil profiles that follow can be aided by the substantially lower pH values of the soil. Soil erosion influences sand content, making it a useful indicator of soil degradation across a variety of land-use types (*Zhou et al., 2019*). Therefore, the SOC pattern can be changed by any factor that affects soil quality. Similarly, compacted soils are more resistant to being broken down by root systems. The activities of roots, aerobic microbes, and animals can all be suppressed by reduced aeration in compacted soils (*Kuzyakov, 2002*; *Tian et al., 2019*). Compacted soils have lower water infiltration rates and may develop anaerobic conditions in puddled areas.

Climate factors such as mean annual temperature (MAT) and mean annual precipitation (MAP) are widely regarded as the most important determinants of soil organic carbon accumulation at large geographical scales and over time because they regulate inputs from litter production and outputs from decomposition. Likewise, NPP and OM input into the soil are both influenced by climate. Microbial activity in soil is also an essential factor in the climate's influence on SOC storage (*i.e.,* the SOM decomposition rate) (*Jackson et al., 2005*; *Jobbágy & Jackson, 2000*). Our meta-analysis showed that soil C sequestration increases with precipitation and decreases with temperature (Fig. S5) because the increasing precipitation stimulates vegetation growth and productivity as a result of an increased SOC sequestration rate (*Chen et al., 2022*; *Liu, Shao & Wang, 2011*). Our findings were consistent with global trends (*Jobbágy & Jackson, 2000*). Previous studies have demonstrated that SOC sequestration varied significantly with vegetation types. For example, during an assessment

of the patterns and environmental controls of soil organic carbon, according to *Chen et al. (2016)*, the average SOC sequestration in P. *crassifolia* forest and grassland at a soil depth of 0–50 cm was 23.68 and 10.11 kg m$^{-2}$ respectively.

Our findings showed that SOC was low at 250 mm compared to 500 mm, indicating that SOC accumulation in Chinese forests is mainly restricted by water availability. Namely, increasing precipitation may reduce SOC decomposition by reducing microbial activity because high soil water availability may limit microorganisms' access to oxygen and hinder SOC decomposition (*An et al., 2022*; *Poeplau et al., 2018*; *Yu & Zhuang, 2020*). In addition, SOC sequestration decreases with temperature due to SOM decomposition temperature dependence. Increased soil temperature accelerates the decomposition of soil organic matter, which decreases the amount of OM stored in soils by transferring soil C into the atmosphere (*Davidson et al., 2008*; *Ngaba, Ma & Hu, 2020*). Hence, lower temperatures could result in reduced SOC breakdown. SOC sequestrations tend to accumulate along the temperature gradient, according to *Ge et al. (2020)*, since the temperature-induced increase in soil carbon decomposition is substantially smaller than the increase in vegetation production.

## Implications and uncertainties of the study

The balance between input and production of C in the soil is critical to soil C sequestration. In the temperate and subtropical forests of China, the response to nitrogen deposition was estimated to be between 5.9 and 10.7 kg C kg$^{-1}$ N. The magnitude of other forest ecosystem types has been reported in previous studies. *De Vries, Du & Butterbach-Bahl (2014)*, in tropical forests, they reported that soil C sequestration was 5.4 kg C kg$^{-1}$ N, which is in line with the value range suggested by *Hermundsdottir & Aspelund (2021)* of 8.6 to 10.5 kg C kg$^{-1}$ N. Soil C sequestration varied between 3 and 25 kg C kg$^{-1}$ N in tropical forests (*Frey et al., 2014*; *Janssens et al., 2010*), and between 10 kg C kg$^{-1}$ N in an N-limited forest (*Maaroufi et al., 2015*). The study conducted by *Du & De Vries (2018)* will bring more clarity by specifying that N deposition leads to a larger C sink in the temperate forest (0.11 Pg C yr$^{-1}$) than in the tropical forest (0.08 Pg C yr$^{-1}$), which is in line with our findings.

Uncertainties might remain in our study when determining the carbon sequestration rate due to the lack of information on the fraction of N retention in Chinese forest ecosystems. Indeed, no study to date has determined this parameter, which led us to use the values obtained in Europe (50% and 15% for temperate and subtropical forests, respectively) (*De Vries, Du & Butterbach-Bahl, 2014*; *De Vries et al., 2007*). It is well known that biological processes control N retention in forest ecosystems, but these processes vary from one site to another. Therefore, although its values were determined in the same forest types, several factors (such as the nitrogen deposition rate, tree species types, climate conditions, vegetation, and microbial uptake) due to the specific sites conditions and environment could influence N retention rate. Moreover, heterotrophic demand for N, soil properties, nonbiological retention of N, land-use change, or plant community can also significantly impact N retention, for instance (*Barrett & Burke, 2002*; *Johnson, Cheng & Burke, 2000*; *Johnson, 1992*; *Silver et al., 2005*; *Templer et al., 2008*). In addition, experimental results

on root respiration, microbial respiration, and SIC were relatively scarce and shed limited insights on the actual N deposition effect in belowground C forest ecosystems. Nevertheless, our results represent the best knowledge we can derive and give an overview of carbon sequestration in response to the nitrogen deposition in Chinese forest ecosystems.

## CONCLUSIONS

Soil carbon sequestration varied significantly across the Chinese forests, with different levels of N form, N addition, NUE, and N application time ranging from 6.8 to 11.2 kg C kg$^{-1}$ N. Soil C sequestration response was significantly higher in temperate forests (10.6:8.37–13.68 kg C kg$^{-1}$ N) compared to subtropical forests (5.87:4.83–7.16 kg C kg$^{-1}$ N). In addition, C sequestration was more sensitive to $NH_4NO_3$, low N addition rate (<30 kg C ha$^{-1}$ yr$^{-1}$), high NUE rate (>7 kg C ha$^{-1}$ yr$^{-1}$), and short-term N application (<1 year). The relative influence showed that nitrogen availability has the strongest impact on the SOC, followed by DOC > MAP > MAT > clay content > pH > sand content > silt content. All things considered, the most promising strategy for carbon sequestration in N-limited forest soils is the incorporation of organic inputs in alongside with N deposition.

### Funding

This work was supported by the National Natural Science Foundation of China (No. 41671295), the National Key R & D Program of China (No. 2017YFD0200106), and the 111 Project (No. B12007). The funders had no role in study design, data collection and analysis, decision to publish, or preparation of the manuscript.

### Grant Disclosures

The following grant information was disclosed by the authors:
National Natural Science Foundation of China: 41671295.
National Key R & D Program of China: 2017YFD0200106.
111 Project: B12007.

### Competing Interests

The authors declare there are no competing interests.

### Author Contributions

- Mbezele Junior Yannick Ngaba conceived and designed the experiments, performed the experiments, analyzed the data, prepared figures and/or tables, authored or reviewed drafts of the article, and approved the final draft.
- Yves Uwiragiye conceived and designed the experiments, authored or reviewed drafts of the article, and approved the final draft.
- Jianbin Zhou conceived and designed the experiments, authored or reviewed drafts of the article, and approved the final draft.

## Data Availability

The raw data are available in the Supplemental File.

## Supplemental Information

Supplemental information for this article can be found online at http://dx.doi.org/10.7717/peerj.14694#supplemental-information.

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
