# Peer review of "Patterns and controlling factors of soil carbon sequestration in nitrogen-limited and -rich forests in China—a meta-analysis"

_PeerJ, doi:10.7717/peerj.14694_

## Round 0.1 · original submission · Major Revisions

After checking the manuscript and reports from expert reviewers, I came to conclude that your manuscript needs major revisions. Please revise and resubmit.

Reviewer 1 ·

Basic reporting

Please see my comments at "Additional comments".

Experimental design

Please see my comments at "Additional comments".

Validity of the findings

Please see my comments at "Additional comments".

Additional comments

peerj -77218: This paper is interesting. But, some issues must be improved.
1) In the abstract, “(P <0.05)” is unnecessary to present. You can present in the wording as “significant”.
2) Lines 53-54: I understand that N deposition is the main issue for this study. But, “Several biological factors influence the processes of soil organic carbon sequestration and decomposition…”. You must provide more details by mentioning to other factors, then convince why N deposition is important. For the other factors influencing SOC, please see these papers. [2012. Carbon quality and the temperature sensitivity of soil organic carbon decomposition in a tallgrass prairie. Soil Biol. Biochem, 50, 142–148.] [2022. Soil organic carbon and soil erodibility response to various land-use changes in northern Thailand. Catena. 219; 106595.] [2005. Carbon stocks in Swiss agricultural soils predicted by land-use, soil characteristics, and altitude. Agric. Ecosyst. Environ., 105, 255–266.] [2020. Factors Controlling Soil Organic Carbon Sequestration of Highland Agricultural Areas in the Mae Chaem Basin, Northern Thailand. Agronomy, 10, 305.] [2017. Spatial distribution of soil organic carbon and its influencing factors under the condition of ecological construction in a hilly-gully watershed of the Loess Plateau, China. Geoderma, 296, 10–17.]
3) Line 130: “Data was collected as previously described in Ngaba et al. (2022).” is unclear for this paper. It is unfair for readers to find the details in Ngaba et al. (2022). You must provide the important details in this paper by referring to Ngaba et al. (2022).
4) Lines 130-140: How many papers were found? Then, how many papers were used and excluded? These details must be mentioned.
5) Discussion section needs more explanation.
5.1) Please explain negative correlation between SOC with bulk density, pH, and sand content.
5.2) Based on your analysis, what is the direction of future research for more understanding the patterns and controlling factors of SOC?
6) The text in the graph and resolution of graph must be clearly presented, especially Figures 1-3 should be improved.

Reviewer 2 ·

Basic reporting

L23: Please provide the full form of CO2
L24: What is the approach used?
L30: It would be better to present the ionic form of the fertilizer (-NH4+ and -NO3-)
L32: Place comma ….. rainfall, fertilizer, and carbon ratio.
L33: Is it fertilizer rates and carbon-nitrogen ratio?
L33. It would be better to put SOC in the parenthesis in L22.
L37-38: The sentence is not clear. Mean annual precipitation suddenly appeared here.
L68: Please include references showing the relation of N deposition, increase in NPP, and decrease of SOC decomposition. Correctly place the SOC in the parenthesis.
L87. Would you please provide reference and evidence to your claim that what factors have increased the N deposition from 13.2 to 21.2 Kg N ha-1 yr-1 in a two-decade period?
L92-99: This is what you should explain at the very beginning of the intro section. This will help to set a scene for your research problem background and definition of the key terms.

Figure 2. Would you please keep the information on orange, green, and blue colours as a legend in the figure rather than explaining them in the caption? So, the figure is complete in itself.

Figure 5 Caption is not complete.

Table S1: What is N dep.? Explain it in the caption. All of the units are in Kg N ha-1 yr-1, so better to remove this column and provide the unit in the column of N dep.

Experimental design

L130: Each of the manuscripts/scientific articles should stand independently. We would like to suggest paraphrasing the data collection approach used in Ngaba et al (2022) in this paper too and later citing Ngaba et al for the details.
L131. Your methodology is a bit confusing. Is this a work resulting from the combined data collection approach i.e. primary (own research sites) and secondary data (literature database)?
L132: Why only between 2000 to 2022, why not earlier than 2000? It would be better to provide the justification for the study period selection.
L139-140. This is an incomplete sentence (verb is missing), please revise it.
L142: So, this is meta-analysis. It would be better to include this in the paper title. Please present the PRISMA flow chart before section 2.2. How many articles you have retrieved from different databases? What are your inclusion-exclusion criteria?
L144: Is METAWIN software or a model?
L145. What statistical approaches have you adopted? Please write the formula and cite the references.

Validity of the findings

L149: SOC sequestration could be the right term rather than SOC concentration.
L172-180: Interesting results. Would you please elaborate and rearrange for clarity? It would be better to re-arrange your findings - what N rate to what duration is effective for C sequestration and how is this differentiated by forest type and climate?
L182-192: So, you have done correlation and relative influence analysis (a part of multiple regression) to analyse SOC controlling factors. Have you checked the multi-collinearity? Please mention this test in your section 2.2.
L203 and L210: There is a logical leap in the discussion. In L203 you argued higher N rate resulted in higher C sequestration and in L210 you again argued higher N accumulation (deposition) resulted in low C sequestration. In both cases, providing their cut-off value is important so that to what extent N addition and deposition govern C sequestration rather than using comparative terms (higher and lower).
L245: Is there any specific sustainable forest management practices recommendation from your findings?
L342: It is not advised to start a sentence with an abbreviation. Please provide the full form of MAP.
L344-345: The concluding statement of the manuscript is vague and generic. Please revise it and make it specific to your findings.

Reviewer 3 ·

Basic reporting

This manuscript discusses the influence of nitrogen enrichment on carbon sequestration in forest soils and its controlling factors in China.This is a clear research manuscript with an acceptable storyline, it is well structured and discussed. There is sufficient information in the introduction. However, there are still some minor problems in the manuscript.

Experimental design

Data obtained from published articles.
Overall, I felt it was good.

Validity of the findings

This is a clear research manuscript with an acceptable storyline, it is well structured and discussed.

Additional comments

1.In the abstract “The relative influence indicated that nitrogen availability strongly impacts the SOC, followed by DOC concentration and water availability.” But in line 191 “ the relative influence analysis indicated that N availability (40%) is one of the major factors controlling C accumulation, followed by DOC> MAP> MAT> clay> pH> sand> silt” . I think water availability is not equal to MAP, they are different.
2.L128, It is suggested to add a section to describe the summary of the study area, including temperature, rainfall, forest type, area, distribution, etc. I think this is very necessary, especially for readers who do not know China.
3.L142, It is suggested to add some content about the meta-analysis method, although this method is more known and used by many people. For example, you should add information about the advantages or disadvantages of this method, the parameters that need to be set in the software, and so on. At present, it is too simple.
4.L151, The words in Figure 1 can't be seen clearly. It is recommended to reproduce and modify them to make them clearer.
5.L162 “MAP” ?? is “mean annual precipitation”? Please use the full name in the first place.
6.L183 Figure 4 has the same problem as Figure 1. It is recommended to revise it again.
7.L190 “MAT” Please use the full name in the first place.
8.L194 It is suggested to add a section on the possible uncertainty in this manuscript. Because the soil data of this manuscript is based on the published articles, their research had different research purposes, different sampling depths, sampling schemes and different analytical methods. You need to discuss and explain the possible uncertainties in this study.
9.L204 There are both Figure and Fig. in the text of the manuscript. Please unify.

---

## Round 0.2 · Minor Revisions

Thanks for revising the manuscript, it has improved a lot since its previous version. However, there still remain some minor issues as pointed out by Reviewer #2, that need to be addressed. Please revise and re-submit.

Reviewer 1 ·

Basic reporting

no comment

Experimental design

no comment

Validity of the findings

no comment

Additional comments

no comment

Reviewer 2 ·

Basic reporting

Title: Do not use a full stop, add a hyphen before "A meta-analysis"
Patterns and controlling factors of soil organic carbon sequestration in nitrogen-limited and -rich forests in China - A meta-analysis

Still, there is inconsistency in the use of abbreviated terms. For example, see Line 9 NUE is not defined earlier, please provide the full form and keep NUE in the parenthesis. In line 16 the same term is expressed differently like "soil N use efficiency", please replace this with NUE. Similarly, check other abbreviations (C, SOC, N, TN etc.) used throughout the text during proofreading. Check L162, 172 etc. how the term NUE is inconsistently expressed.

L134: Move this sentence "Data was .....Ngaba et al" to the last of this paragraph.

Experimental design

All good

Validity of the findings

All good

Additional comments

It will be ready for publication after a few changes to terminology and a few syntax corrections. We would like to thank and appreciate the authors for their extensive amount of work.

Reviewer 3 ·

Basic reporting

This manuscript discusses the influence of nitrogen enrichment on carbon sequestration in forest soils and its controlling factors in China. I appreciate the authors including the comments into current revision. In my opinion, the present manuscript has been improved and could be accepted for publications now.

Experimental design

It's OK.

Validity of the findings

It's OK.

Additional comments

It's OK.

---

## Round 0.3 · accepted · Accept

Thanks for the revision. The manuscript has been accepted.